# Multimodal Web-Based Intervention for Cancer-Related Cognitive Impairment in Breast Cancer Patients: Cog-Stim Feasibility Study Protocol

**DOI:** 10.3390/cancers13194868

**Published:** 2021-09-28

**Authors:** Giulia Binarelli, Marie Lange, Mélanie Dos Santos, Jean-Michel Grellard, Anaïs Lelaidier, Laure Tron, Sophie Lefevre Arbogast, Benedicte Clarisse, Florence Joly

**Affiliations:** 1Clinical Research Department, Centre François Baclesse, 14000 Caen, France; m.lange@baclesse.unicancer.fr (M.L.); m.dossantos@baclesse.unicancer.fr (M.D.S.); JM.GRELLARD@baclesse.unicancer.fr (J.-M.G.); s.lefevre-arbogast@baclesse.unicancer.fr (S.L.A.); clarb@baclesse.unicancer.fr (B.C.); f.joly@baclesse.unicancer.fr (F.J.); 2Interdisciplinary Research Unit for the Prevention and Treatment of Cancers (ANTICIPE), National Institute of Health and Medical Research (INSERM), University of Caen Normandie (UNICAEN), Normandie University, 14000 Caen, France; laure.tron@inserm.fr; 3Cancer and Cognition Platform, Ligue Nationale Contre le Cancer, 14000 Caen, France; 4Northwest Data Center (CTD-CNO), Ligue Nationale Contre le Cancer and French National Cancer Institute (INCa), 14000 Caen, France; a.lelaidier@baclesse.unicancer.fr

**Keywords:** cancer-related cognitive impairment, web-based interventions, multimodal interventions, breast cancer, physical activity, cognitive stimulation

## Abstract

**Simple Summary:**

Cognitive difficulties and their impact on patients’ quality of life are frequently reported by patients treated for breast cancer, who ask for support to improve these difficulties. Cognitive stimulation and physical activity resulted as beneficial for cognitive difficulties, but they are challenging to generalize in hospitals. To overcome this limitation, home-based computerized interventions have been proposed. In this study, the feasibility of a combined intervention of web-based cognitive stimulation and physical activity among breast cancer patients undergoing radiotherapy will be investigated. The overall goal is to develop interventions for cognitive difficulties adapted to supportive care units.

**Abstract:**

Cancer-related cognitive impairment (CRCI) is a frequent side-effect of cancer treatment, with important consequences on patients’ quality of life. Cognitive stimulation and physical activity are the most efficient in improving cognitive impairment, but they are challenging to generalize in hospitals’ routine and to patients’ needs and schedules. Moreover, the added value of a combination of these interventions needs to be more investigated. The Cog-Stim study is an interventional study investigating the feasibility of a web-based multimodal intervention (combining cognitive stimulation and physical activity for the improvement of cognitive complaints among breast-cancer patients currently treated with radiotherapy (*n* = 20). Patients will take part in a 12-week program, proposing two sessions per week of web-based cognitive stimulation (20 min/session with HappyNeuron^®^) and two sessions per week of web-based physical activity (30 min/session with Mooven^®^ platform). Cognitive complaints (FACT-Cog) and objective cognitive functioning (CNS Vital Signs^®^), anxiety, depression (HADS), sleep disorders (ISI) and fatigue (FACIT-Fatigue) will be assessed before and after the intervention. The primary endpoint is the adherence rate to the intervention program. Patients’ satisfaction, reasons for non-attrition and non-adherence to the program will also be assessed. The overall goal of this study is to collect information to develop web-based interventions for cognitive difficulties in supportive care units.

## 1. Introduction

Cancer-related cognitive impairment (CRCI) is one of the most frequent side-effects of cancer and its treatments [1,2,3,4,5]. The impact of chemotherapy on cognition is the most documented in the literature [1]. However, studies show that radiation therapy and hormone therapy can also affect cognition [6,7]. However, 20% to 30% of patients were shown to experiment CRCI even before adjuvant treatment [8]. Furthermore, breast cancer patients had more objective cognitive impairment before any treatment (including breast cancer surgery), compared with healthy controls [9]. Patients treated for cancer frequently report (40–75%) having trouble in remembering, thinking, concentrating or finding the right words [2,10,11]. These symptoms can persist even 10 to 20 years after breast cancer chemotherapy [12,13] and the repercussions are so profound that they perceive themselves as “chemobrain victims” [14]. Indeed, these symptoms impact patients’ quality of life [15,16,17], disrupt their return to work [18] and lead to a decrease in self-confidence at work or in social relationships [15,19,20,21]. For example, in interviews conducted one year after breast cancer treatment, survivors reported deterioration in quality of life and daily functioning due to cognitive difficulties, as well as coping strategies implemented to manage their work and social life [15]. Some patients also reported that cognitive impairment was the most troublesome symptom after treatment. In addition, approximately 6 years after chemotherapy, patients in remission from breast cancer found that cognitive difficulties were frustrating and affected their self-confidence and social relationships [21]. Half of these women reported working harder to complete tasks and using compensatory strategies to complete tasks at work [21]. Because these symptoms can occur before or during cancer-treatment, it seems crucial to intervene during medical treatment, to alleviate their impact on patients’ quality of life and to help them gain self-esteem before returning to work. Moreover, as suggested by Lonkhuizen et al. [22], an early intervention could prevent or delay worsening of CRCI, resulting in a better long-term efficacy. To date, there are still no available and validated guidelines for the management of cognitive complaints. The majority of cancer survivors reporting CRCI (75%) would like to receive support, especially cognitive stimulation (72%) combined or not with physical activity [2].

Different methods have been evaluated and preliminary findings have shown beneficial effects of non-pharmacological approaches [23]. Among these, cognitive stimulation and physical activity are especially promising for the improvement of cognitive complaints and quality-of-life of cancer patients [2,22,24,25,26].

Nevertheless, further high-quality studies are needed to identify the best efficient approach between these interventions. Indeed, to date, the efficacy of either cognitive stimulation or physical activity has been only compared to “standard care” (“standard group” or “wait-list” group only). Therefore, comparative studies are needed to identify the best possible intervention.

Furthermore, the intervention, whatever its modality, has been proposed mostly to survivors a long time after the start of symptoms, and few of these studies evaluated adherence ratings in the different groups of interventions.

The possible cognitive enhancing effects of a combination of physical activity and cognitive stimulation have been suggested by several authors. Indeed, it seems that both interventions can increase neuroplasticity and processes related to cognition and associative learning, but via different neuronal mechanisms [27,28]. Fissler et al. (2013) [28] proposed the model “guided plasticity facilitation” in which they suggest physical activity to be the “plasticity facilitator” because it facilitates synaptic plasticity and neurogenesis, while cognitive training regulates (guides) synapse formation and elimination [29].

This hypothesis has been confirmed by several studies outside oncology research, [27,28,30,31,32,33,34,35] although, in the field of CRCI, to our knowledge, only two small (10 and 28 patients, respectively) studies, with low statistical power, have reported some preliminary results on the potential effect of a computer-based intervention combining cognitive stimulation and physical activity [36,37]. In both studies, the intervention consisted of 12 weeks of computer-based cognitive training; aerobic training, or a combination of both cognitive and aerobic training followed by flexibility (30 min-stretching). In both studies, the control group performed a 30-min flexibility training. The combined intervention was done on a Motion Fitness Brain-Bike, on which it was possible to do cognitive and physical exercises at the same time. The electric bicycle was equipped with a screen on which patients did cognitive exercises while pedaling. In neither study did the combined intervention lead to a significant increase in cognition. Authors suggested that the absence of significant beneficial effects could be a result of the high level of difficulty and stress reported among patients included in the combined intervention group. Results from these studies differ from previous results from studies investigating the efficacy of simultaneous combined intervention in other populations [28,32,38]. It can be suspected that the mode of intervention in these two studies was too overwhelming and stressful for patients so that it could result with beneficial effects on cognition.

The setup of such multimodal interventions in standard supportive care is difficult, due to structural or health system-related barriers (such as cost and lack of trained healthcare providers). Such programs are also not adapted to patients’ schedule and needs (e.g., patients had to return frequently to the care center for the center’s sessions in addition to their doctor’s appointments). E-Health approaches represent one solution to overcome these barriers because it allows home-based and remotely supervised interventions. The potential of such approaches has already been shown in multiple domains of mental health [39,40,41,42] and also for the improvement of quality of life [43] and CRCI in cancer patients [44,45,46,47,48,49,50]. Furthermore, e-Health interventions have been shown to improve patients’ engagement [51] and to promote physical activity [52]. However, patients’ acceptance and easiness in the use of computerized interventions, and their opinion concerning the length of the program and frequency and length of the sessions is still unclear.

In conclusion, while previous studies have enlightened the interest of using a computerized multimodal intervention, combining cognitive stimulation and physical activity, it is imperative to investigate further the feasibility and acceptability of a computerized multimodal intervention, before exploring their efficacy for cognitive improvement in randomized controlled studies. Thus, in this pilot study, we aim to bridge this gap, investigating the feasibility and acceptability of a program proposing a web-based multimodal intervention combining cognitive stimulation and adapted physical activity (APA) to breast cancer patients undergoing radiotherapy who have cognitive complaints.

## 2. Materials and Methods

### 2.1. Study design

The Cog-Stim study is a feasibility interventional study conducted in the Cancer Comprehensive Center of Caen (Caen, France).

Several meta-analysis aiming at evaluating the efficacy of combined interventions reported better efficacy in interventions with less than five sessions per week, with short (<45 min) or medium (45 to 60 min) duration per session [32,38]. Conversely, the length of the intervention was not shown to impact on the efficacy of the combined intervention.

Based on these findings and considering the design of previous studies on combined interventions in oncology [36,37], we designed a 12-week program combining web-based cognitive stimulation (2 sessions per week, 20 min per session) and APA (2 sessions per week, 30 min per session). The program will be proposed to breast cancer patients reporting cognitive complaints and undergoing adjuvant radiotherapy. Participants will be assessed pre-intervention (T0) and post-intervention (T1) (Figure 1).

### 2.2. Objectives

#### 2.2.1. Primary Outcome

The main objective of the Cog-Stim study is to evaluate adherence of breast cancer patients with cognitive complaints undergoing radiotherapy to a 12-weeks intervention combining web-based cognitive stimulation and web-based adapted physical activity. A patient will be considered as adherent to the 12-weeks intervention if 70% of the planned sessions or more are realized. A session will be considered performed if at least 70% of the session has been completed.

#### 2.2.2. Secondary Outcomes

Secondary objectives are:(1)To evaluate the proportion of acceptance (attrition) of the study among breast cancer patients starting adjuvant radiotherapy, and according to previous adjuvant chemotherapy. The number of patients contacted, program rejection and acceptance rates, as well as reasons for rejection will be recorded, along with the presence or absence of prior chemotherapy treatment.(2)To evaluate patients’ satisfaction regarding the proposed intervention program (frequency, duration, content of the sessions, whether the moment to initiate cognitive rehabilitation throughout the oncologic management is appropriate, obstacles to access and use the different software, and the perceived usefulness of the program). This information will be collected using a 13-item questionnaire, developed in our institution for this study.(3)To identify barriers to accessing the program (impossibility or inability to use the computer, no access to internet connections etc.) or to achieve its completion (motives of non-eligibility, non-participation, and drop-out, etc.). This information will be collected and stored by the neuropsychologist throughout the research program;(4)To evaluate exercise intensity and training burden and its possible impact on adherence to the protocol. This information will be collected by the APA specialist and stored on the Mooven App.

These secondary objectives will enable us to identify potential limitations of the program and to consider adjustments to improve and optimize the study design to develop a randomized control trial with web-based multimodal intervention.

Furthermore, the pre-post intervention of cognitive complaints, cognitive objective functioning, anxiety, depression, sleep disorder and fatigue will be assessed.

### 2.3. Participants: Recruitment and Procedure

This feasibility study aims to recruit 20 patients with breast cancer, reporting significant cognitive complaints related to cancer and its treatments and undergoing radiotherapy. Patients will be enrolled while undergoing radiotherapy treatment, in order to assess the feasibility of an early combined intervention during medical care. All patients will give their written informed consent to the study which was approved by the local ethics committee [Ref. 2019-67, Comité de protection des personnes Nord Ouest III, France]. This trial is registered as ID RCB 2019-A02500-57, clinical trial NCT04213365 (https://clinicaltrials.gov/ct2/show/NCT04213365?term=cog+stim&draw=2&rank=1, accessed on 27 Janunary 2021).

### 2.4. Inclusion Criteria

Patients will be included if they meet all the following criteria: (1) breast cancer patient aged 18 or more; (2) undergoing adjuvant radiotherapy, independently of previous chemotherapy but before any hormone therapy treatment; (3) reporting cognitive complaints with significant impact on their quality of life (quality of life subscale of the Functional Assessment of Cancer Therapy Cognitive Scale (FACT-Cog) score ≤the 10th percentile, according to age [53]); (4) absence of any major cognitive dysfunction that could prevent the patient from achieving the neuropsychological testing (threshold based on Montreal Cognitive Assessment (MoCA) score [54], according for patient’s age and educational level, GRECOGVASC normative data [55]); (5) absence of personality disorder or any other known progressive psychiatric pathology (e.g., schizophrenia); (6) absence of neurological antecedent (neurological sequelae of brain injury, stroke with loss of consciousness >30 min, multiple sclerosis, epilepsy, neurodegenerative pathology, etc.); (7) patient having completed at least an educational level of “end of primary school”; (8) patient with an access to a functional laptop/computer with a keyboard, headphones or speakers, internet connection and an e-mail account-being able to use those tools alone; (9) fluent in French; (10) having signed the informed consent to participate in the study.

Participants who fulfil any of the following criteria will not be included in this study: (1) excessive alcohol intake or drug use (according to medical record since frequency of alcohol and drug intake is collected in routine hospital practice during medical management of patients); (2) major visual and/or hearing deficit; (3) patient who might not be able to complete neuropsychological testing; (4) medical contraindication to adapted physical activity; (5) refusal to participate; (6) patient deprived of liberty or under guardianship; (7) patient who might not be able to participate due to geographic, social or psychopathological reasons (patients living in rural areas with a poor internet connection, absence of time, poor physical or mental conditions etc.).

### 2.5. Assessments

The baseline assessment (T0), before participating in the intervention program, will be performed by a neuropsychologist in the Cancer Comprehensive center and the follow-up assessment after the 3-month intervention program (T1), on-site or at home, based on the patient’s preference. The MoCA test will be performed at baseline (inclusion criteria).

Objective cognitive function will be assessed at T0 and T1 with the computerized neurocognitive test battery CNS Vital Sign [56]. This battery uses computerized forms of traditional tests such as symbol digit coding, the Stroop test and finger tapping and has also the capacity to automatically quantify “speed factor” via multiple parameters such as reaction time, psychomotor speed and processing speed. Test-retest reliability, concurrent validity with traditional tests and discriminant validity was assessed and showed similar characteristics to the traditional neuropsychological tests [56]. In this study, we will use the 7 main tests of this battery (30 min):Verbal Memory (VBM), which investigates word recognition and words memorization ability (immediate and delayed recall);Visual Memory (VIM), which investigates recognition and memorization of geometric shapes (immediate and delayed recall);Finger Tapping (FTT), for motor speed and fine motor control assessment;Symbol Digit (SDC) to investigate information processing speed, complex attention, visual-perceptual speed and complex information processing speed-accuracy;Stroop test (ST) for simple and complex reaction time, inhibition/disinhibition abilities, processing speed and frontal or executive skills assessment;Shifting Attention (SAT) to assess executive functions, decision making and reaction time;Continuous Performance (CPT) for sustained attention, impulsivity and choice reaction time.

Subjective cognitive complaints will be evaluated at T0 and T1 through the self-report questionnaire FACT-Cog, which is validated in French, with normative data, and has 4 subscales: Perceived Cognitive Impairments (PCI), Impact on Quality of Life (QoL), Comments from Others (Oth), and Perceived Cognitive Abilities (PCA) [53,57].

To explore the potential secondary effects of the intervention, the following will be assessed at T0 and T1: anxiety and depression will be assessed with the Hospital Anxiety and Depression Scale (HADS) [58], fatigue with the Functional Assessment of Chronic Illness Therapy-Fatigue (FACIT-Fatigue) [59], and quality of sleep with the Insomnia Severity Index (ISI) [60]. Patients’ satisfaction will be evaluated at T1 through a 13-items self-report questionnaire developed in our institution for this study. The questionnaire focuses on patients’ general appreciation on the program; on frequency, difficulty, contents and length of physical and cognitive sessions; software’s easiness of use; the time of the proposed study. Each question has 4 possible answers (from not satisfied at all, to very satisfied) (see Appendix A).

Data collection information is summarized in Table 1.

### 2.6. The Multimodal Web-Based Intervention

All participants will be enrolled in a web-home-based multimodal intervention for 12 weeks and will be asked to perform two 20-min sessions of cognitive stimulation per week and two 30-min sessions of APA per week (Figure 1).

After inclusion in the study, participants will meet a neuropsychologist who will introduce and explain to them the 2 online platforms and will give them all the needed materials. A starter kit will be provided at that moment, including a heart rate monitor to adapt the activity during the APA sessions as well as a webcam (if necessary) for the video-conference sessions. This meeting will allow participants to familiarize themselves with all instruments of the study and with the person with whom they will be in contact during the study.

Furthermore, an APA specialist will conduct an initial physical activity interview (45-to-60-min) during which he/she will present the program to the patient and check the patient’s availability for APA sessions and medical contraindications to adapt the sessions according to the needs and constraints of the patients.

#### 2.6.1. Cognitive Stimulation—HappyNeuron Platform^®^

Cognitive stimulation will be remotely provided with the HAPPYNeuron^®^ PRESCO online program (https://www.happyneuronpro.com/, accessed on 27 September 2021) This software has been developed by a neurologist and it is available in 11 languages allowing a comparison of results between international studies. It has already been used in several clinical studies, mainly in the psychiatric field, and in a study with breast cancer patients [47]. This home-based program trains 12 cognitive domains, especially attention, memory, executive functions and processing speed—which are the most impaired after cancer treatments [2] and included 41 different exercises and 9 difficulty levels. This program is particularly appropriate for an interventional study because it allows a standardized intervention targeting a wide range of cognitive domains impaired by cancer treatments (Table 2). Moreover, its user-friendly and intuitive interface, along with its ludic component, may help maintaining patients’ motivation and their adherence to the program.

Participants will have access to the training session through a link sent by e-mail and will have access to it until they receive another mail with the next planned session. All patients will start with the first difficulty’s level, which will increase accordingly with their performance during exercises. Patients will reach the next level after two successful repetitions of the exercise at the given level. Instructions and demonstrations precede each exercise as well as a test to verify if instructions have been well understood. Automatic feedback is generated after each exercise to congratulate patients or to encourage them to persevere if they fail. When starting a new session, patients will carry on at the level they stopped during the previous session.

The neuropsychologist will have access to patients’ results to estimate the evolution of their performance and to verify the frequency with which sessions are being completed. The neuropsychologist also will perform a weekly check for patients’ achievement to the cognitive stimulation program and in case of non-achievement of two following sessions, he/she will contact the patient to verify reasons of non-achievement.

#### 2.6.2. Adapted Physical Activity Training—Mooven^®^ Platform

Participants will access adapted physical activity sessions through the online platform “Mooven^®^” (https://mooven.app/, accessed on 27 September 2021). It is currently involved in more than 15 projects, including a trial with cancer patients.

Participants will schedule two 30-min sessions per week with an APA specialist who will remotely supervise the session through a video-conference system. APA sessions will be standardized and will consist of a warm-up (5 min), endurance/cardio or muscle-strengthening activities (20 min) and then stretching (5 min), as recommended by the French National Cancer Institute [61]. The content of the sessions will be adapted to breast cancer patients [62,63] and if necessary, to their constraints and medical contraindications. Patients’ heartbeat will be constantly monitored using a wrist heart rate monitor and data will be updated in patients’ profile. Furthermore, patients’ rating of perceived exertion will be measured by the Borg rating of the perceived exertion scale [64]. This scale ranges from 6 (no exertion at all) to 20 (maximal exertion).

Patients will have access to their calendar with the next planned video-conference sessions. In case of patient’s inability to attend the video call with the APA specialist, the latter will submit a description of the session to be carried out in autonomy (video supports or sheets) and will contact them after the session to collect the difficulties encountered by the patients through a phone-administered questionnaire. The APA specialist will also report reasons for non-achievement if these do occur.

### 2.7. Statistical Analysis

For the pilot/feasibility study, the inclusion of 20 patients is needed to meet the main objective of evaluating adherence of breast cancer patients with cognitive complaints undergoing radiotherapy to a 12-weeks intervention combining web-based cognitive stimulation and web-based adapted physical activity. A sample size of 20 patients to be enrolled (including the augmentation to anticipate a maximum of 10% of patients with non-evaluable data) will allow estimation with a 95% confidence interval of width +/− 20% an adherence rate of at least 70%.

We expect to reach an adherence rate of 70% (to the APA sessions and the cognitive training sessions) in the present study. Completed sessions and duration will be checked by the neuropsychologist from the patients’ statistics in the HAPPYNeuron^®^ platform regarding cognitive training. Regarding APA sessions, the APA specialist will collect the information directly from patients’ feedback after each session and will confront this self-reported information to those collected by the pedometer (objective measure).

Analysis of non-eligibility motives and factors associated with participation (through data collected by the neuropsychologist at baseline and the coach during the follow-up) as well as analysis of the satisfaction questionnaire will enable us to raise potential limitations of the program such as:social barriers (patients living in remote areas with a poor internet connection, patients with limited access to computer material or low skills in informatics, etc., resulting in selection bias of more affluent women and social inequalities in the access to the program),technological barriers (patients with limited skills in informatics, inappropriate material),or availability barriers (already in an APA or cognitive training program, program too demanding/exhausting, not available enough to complete it, not the good time during cancer therapeutic management to propose cognitive management, etc.).

These key elements will be crucial to identify potential improvements and adjustments of the intervention design for the next steps to develop a randomized study.

Data analysis will be based on the data obtained from the total population (“intent to treat” analysis) enrolled in the protocol. Exploratory data analyses will provide, for qualitative variables, the frequencies and their exact 95% confidence interval (binomial law); for quantitative variables, the mean, the standard deviation of the mean, the median and the quartiles. The demographic, clinical and biological characteristics of the patients will be described using the “intention-to-treat” population. Whenever possible, non-parametric tests will be used. The adherence of breast cancer patients to cognitive stimulation sessions coupled with adapted physical activity will be described in detail as well as the reasons for non-acceptance of the study.

## 3. Discussion

Breast Cancer patients are actively demanding interventions to improve their cancer-related cognitive impairment [14,65]. Nevertheless, without available and validated guidelines for the management of cognitive complaints, only 30% of patients receive the help demanded for their cognitive impairment [66]. While the beneficial effects of both cognitive stimulation and physical activity have been proved [2,23,24,25,26], many barriers limit the setup of such interventions in supportive care units in hospitals. Web-based interventions have been used to reduce some of these barriers, including the cost of interventions, while increasing convenience for patients, also reaching patients who are normally isolated and cannot benefit from these interventions [67] and have resulted as efficient in the improvement of CRCI [44,45,46,47,48,49,50]. Nevertheless, little is known about patients’ preferences in terms of length of the program, length, and frequency of the sessions. This pilot study aims to bridge this gap, collecting patients’ opinion on this topic. Thereafter, based on the results from this pilot study, a multisite randomized controlled study will be conducted, to identify the best computerized intervention for the improvement of CRCI.

The web-based/home-based nature of the Cog-Stim study program will allow a generalization of this intervention for all cancer patients with cognitive complaints. This will provide better support for cancer patients with cognitive difficulties by helping them go back more smoothly to daily living activities, social interactions and work, thus improving their quality of life during and after cancer treatments. The value of such interventions has also increased with the COVID-19 outbreak, during which, thanks to web-based approaches, it has been possible to provide remote support.

## 4. Conclusions

The efficacy of web-based cognitive stimulation and physical activity has been demonstrated, but the added value of a combination of these interventions needs to be more investigated. This study will assess the acceptability and feasibility of such combined intervention using web-based platforms in breast cancer patients with cognitive complaints. The data collected from the Cog-Stim study will be the base for the development of a personalized combined intervention to help patients to cope with CRCI.

## Figures and Tables

**Figure 1 cancers-13-04868-f001:**
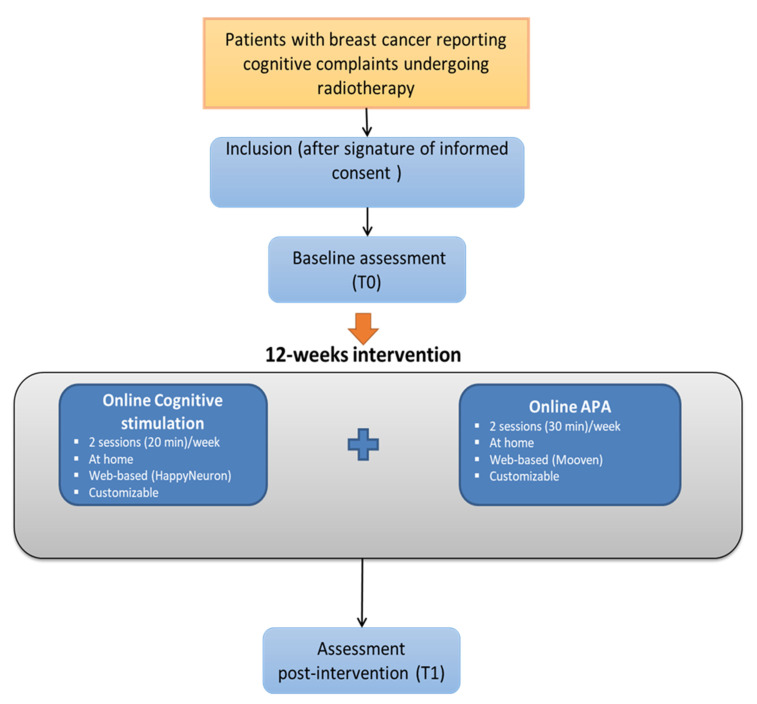
Study flowchart. APA: adapted physical activity.

**Table 1 cancers-13-04868-t001:** Details concerning data collection.

Data Collection	Before Inclusion	Baseline Assessment (T0)	Assessment Post-Intervention (T1)
Signature of Informed Consent	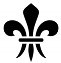		
Clinical ExaminationIncluding medical history ECOG and vital signs	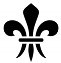		
Cognitive Assessment AND Quality of Life Questionnaires			
FACT-COG	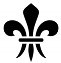		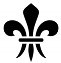
MoCA	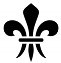		
CNS Vital signs battery		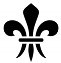	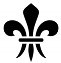
HADS, FACIT-F, ISI		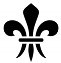	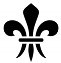
Patient’s Satisfaction			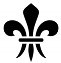

ECOG: Scale of Performance Status; FACT-COG: Functional Assessment of Cancer Therapy—Cognitive Function; MoCA: Montreal Cognitive Asessment; HADS: Hospital Anxiety and Depresion Scale; FACIT-F: The Functional Assessment of Chronic Illness Therapy—Fatigue; ISI: Insomnia Severity Index.

**Table 2 cancers-13-04868-t002:** HappyNeuron^®^ exercises used for cognitive stimulation.

Cognitive Domains Trained	Exercises
Memory (Verbal and Visual)	Words, Where are you? Elephant Memory Shapes and Colours Heraldry Displaced Characters Displaced Images N-Back Around the World in 80 trips I Remember You! Restaurant An American in Paris Find Your Way! Chunking Objects, Where are You?
Executive functions	Towers of Hanoi
	Basketball in New-York
	Hurry for Change!
Attention	Pay Attention!
	Private Eye!
	Sound Check!
	Ancient Writing
Information processing speed	Two-Timing
	Under Pressure
	Gulf Stream
	Catch the Ladybug!
Language	Split words
	Embroidery
	Secret files
	Speak Your Mind!
	Decipher
	Writing in the Stars
	This Story is Full of Blanks!
	Which One is Alike?
Logic	The Right Count
	Countdown
	Ready, Steady, Count!
visuospatial abilities	Sleight of Hands
	Entangled Figures
	Point of View
	Turn Around and Around

## Data Availability

The data presented in this study are available on request from the corresponding author. The data are not publicly available accordingly to good clinical practice (GCP) guidelines.

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
