# Peer review of "Multimodal Web-Based Intervention for Cancer-Related Cognitive Impairment in Breast Cancer Patients: Cog-Stim Feasibility Study Protocol"

_cancers, 2021, doi:10.3390/cancers13194868_

Round 1

Reviewer 1 Report

The revisions to the paper have added greater clarity to the protocol, which will be helpful for other researchers interested in this project.  Some minor issues continue to warrant attention:

  1. The rationale for focusing on patients undergoing radiotherapy remains only broadly addressed in the manuscript. It is noted in the authors choice was based on the desire to provide intervention during the period of radiotherapy, which is standard therapy for breast cancer patients. However, as written, the background does not provide a description of what is known about how radiotherapy can impact cognitive functioning. Rather, the background emphasizes the role of chemotherapy.
  2. Related to the point above, the authors provide a rationale for not including chemotherapy in the inclusion criteria, as described in lines 180-182. However, the meaning of this sentence is somewhat confusing and could perhaps be omitted altogether if a stronger justification on focusing on radiotherapy is provided.   
  3. Line 243 - I believe this should more accurately read "To explore the potential secondary effects of intervention, the following will be assessed at T0 and T1:.... ".
  4. Table 1 - It is unclear what is meant to be communicated with the empty cells in this table.  Perhaps there were meant to be checkmarks indicating data collected at each time point, that have not come through in the reviewer version?

Author Response

Point 1: The rationale for focusing on patients undergoing radiotherapy remains only broadly addressed in the manuscript. It is noted in the authors choice was based on the desire to provide intervention during the period of radiotherapy, which is standard therapy for breast cancer patients. However, as written, the background does not provide a description of what is known about how radiotherapy can impact cognitive functioning. Rather, the background emphasizes the role of chemotherapy

Response 1: As suggested, the introduction has been completed with some additional information on the potential impact of various cancer treatments on cognition: a description of available data on how cognitive functioning is impaired by chemotherapy, as well as radiotherapy or hormonotherapy is nowadays provided, associated with two additional references.

Point 2: Related to the point above, the authors provide a rationale for not including chemotherapy in the inclusion criteria, as described in lines 180-182. However, the meaning of this sentence is somewhat confusing and could perhaps be omitted altogether if a stronger justification on focusing on radiotherapy is provided.

Response 2: We agree with the reviewer: as the introduction section has been modified as above mentioned, it appears relevant to delete the sentence in the “Participants: recruitment and procedure” section. 

Point 3: Line 243 - I believe this should more accurately read "To explore the potential secondary effects of intervention, the following will be assessed at T0 and T1:.... "

Response 3: We agree with the reviewer, and the sentence has been rephrased as suggested. 

Point 4: Table 1 - It is unclear what is meant to be communicated with the empty cells in this table.  Perhaps there were meant to be checkmarks indicating data collected at each time point, that have not come through in the reviewer version?

Response 4: There were checkmarks indicating data collected at each time point, but for unknown reasons in the pdf version the symbols have not come through. In this version they should be visible. 

Reviewer 2 Report

Thank you very much for inviting me to review this protocol. This is a well-designed study to develop and evaluate a home-based computerized intervention combining cognitive stimulation and physical activity for breast cancer patients with cognitive difficulties.  The study background, knowledge gap and research design were well presented. The outcomes, both primary and secondary, allow for evaluation of barriers and facilitators in the implementation of the intervention program. The use of a HAPPYneuron and Mooven for the multimodal components of the program show feasibility and acceptability for cancer patients. Overall, I think the protocol is thorough and explains to any reader the details of the intervention and can be published as it stands.

Author Response

We would like to thank the reviewer for the postives comments and the time he has spent reading our article.